# Test accuracy of faecal calprotectin for inflammatory bowel disease in UK primary care: a retrospective cohort study of the THIN data

Karoline Freeman ,[1] Sian Taylor-Phillips ,[1] Brian H Willis,[2] Ronan Ryan,[2] Aileen Clarke[1]

► Prepublication history and supplemental material for this paper is available online. To view these files, please visit the journal online (http://dx.doi.org/10.1136/bmjopen-2020-044177).

[1]Warwick Medical School, University of Warwick, Coventry, UK
[2]Institute of Applied Health Research, University of Birmingham, Birmingham, UK

**Correspondence to**
Karoline Freeman;
K.Freeman@warwick.ac.uk

## ABSTRACT

**Objective** To estimate the test accuracy of faecal calprotectin (FC) for inflammatory bowel disease (IBD) in the primary care setting using routine electronic health records.

**Design** Retrospective cohort test accuracy study.

**Setting** UK primary care.

**Participants** 5970 patients (≥18 years) without a previous IBD diagnosis and with a first FC test between 1 January 2006 and 31 December 2016. We excluded multiple tests and tests without numeric results in units of µg/g.

**Intervention** FC testing for the diagnosis of IBD. Disease status was confirmed by a recorded diagnostic code and/or a drug code of an IBD-specific medication at three time points after the FC test date.

**Main outcome measures** Sensitivity, specificity, and positive and negative predictive values for the differential of IBD versus non-IBD and IBD versus irritable bowel syndrome (IBS) at the 50 and 100 µg/g thresholds.

**Results** 5970 patients met the inclusion criteria and had at least 6 months of follow-up data after FC testing. 1897 had an IBS diagnosis, 208 had an IBD diagnosis, 31 had a colorectal cancer diagnosis, 80 had more than one diagnosis and 3754 had no subsequent diagnosis. Sensitivity, specificity, and positive and negative predictive values were 92.9% (88.6% to 95.6%), 61.5% (60.2% to 62.7%), 8.1% (7.1% to 9.2%) and 99.6% (99.3% to 99.7%), respectively, at the threshold of 50 µg/g. Raising the threshold to 100 µg/g missed less than 7% additional IBD cases. Longer follow-up had no effect on test accuracy. Overall, uncertainty was greater for specificity than sensitivity. General practitioners' (GPs') referral decisions did not follow the anticipated clinical pathways in national guidance.

**Conclusions** GPs can be confident in excluding IBD on the basis of a negative FC test in a population with low pretest risk but should interpret a positive test with caution. The applicability of national guidance to general practice needs to be improved.

## Strengths and limitations of this study

► The sample size in our study is likely to be larger than the usual cross-sectional studies used to evaluate a test, which is useful for inflammatory bowel disease (IBD) which has a low prevalence.

► We applied strict eligibility criteria to ensure the reliability of reported test accuracy measures and explored the effects of different follow-up times on the definition of IBD.

► We used a very comprehensive and sensitive list of Read codes and drug codes (48 codes) for the identification of IBD.

► The exclusion of close to half of the faecal calprotectin tests without a numerical test result raises concerns over the generalisability of the findings; however, findings are in line with published estimates.

► A coded IBD diagnosis or prescription is a proxy reference standard for a composite of different reference tests used in clinical practice reflecting a pragmatic approach to test accuracy.

reported prevalence values in North America (568 per 100 000 persons) and Europe (827 per 100 000 persons).[1] UK prevalence estimates were 970/100 000 in 2017.[2] It often presents with non-specific symptoms such as abdominal pain and diarrhoea, making it difficult to differentiate from irritable bowel syndrome (IBS), which is far more prevalent in the primary care setting. This also affects the numbers and composition of referrals to secondary care, with a higher proportion of patients being diagnosed in secondary care with IBS than is desirable. It is against this background that the non-invasive faecal calprotectin (FC) test has been introduced as a test for IBD in primary care in the UK. FC is a small calcium-binding protein of the S100 family which is predominantly derived from neutrophils during an immune response. It is associated with inflammation and can be

## INTRODUCTION

Inflammatory bowel disease (IBD) is a chronic, progressive disorder caused by inflammation of the gastrointestinal tract. The prevalence of IBD is rising worldwide, with the highest

quantified in stool samples.[3] Therefore, it is more specific than serum inflammatory markers in the detection of gastrointestinal inflammation.[3]

The FC test is recommended by gastroenterological societies across the globe for its usefulness in the diagnosis of IBD.[4–7] However, there is no clear guidance on which settings it is considered appropriate. The National Institute for Health and Care Excellence (NICE) recommends the use of FC testing in UK primary care for the differential diagnosis of IBD and IBS when a referral is being considered and colorectal cancer (CRC) is not suspected.[8] A study of routine data from an American health plans database of FC testing for chronic gastrointestinal symptoms in primary care suggests that it is used in routine general practice in at least one other country.[9] The NICE recommendations were based on an evaluation of secondary care test accuracy studies and assumptions that referral decisions would follow FC test results. This resulted in a slow uptake and inconsistent use of FC testing in primary care.[10] While primary care studies exist, we still lack compelling evidence on the test accuracy of FC testing in primary care for the detection of IBD. This is because the primary care evidence is based on small, heterogeneous studies that vary in clinical questions, FC thresholds and definitions of disease.[11] They report a range of sensitivities and specificities with wide CIs and leave uncertainty over which threshold (50 or 100 µg/g) should be adopted in primary care. To determine the utility of FC testing in primary care, a pragmatic study is required that both captures the differential diagnosis between IBD and IBS and has a patient population that is representative of primary care.

Thus, our primary aim was to determine the diagnostic value of FC for IBD in primary care by estimating the sensitivity, specificity and predictive values for two thresholds using routine UK primary care data from The Health Improvement Network (THIN).

## METHODS

### Data source

We undertook a test accuracy study of FC testing for IBD using data from the THIN. THIN is a UK longitudinal primary care database of over 14 million patients from more than 670 general practitioner (GP) practices covering about 6% of the UK population in 2015.[12] It has been found to be broadly representative of the UK population.[13] Patients' medical diagnoses, symptoms, laboratory tests, referrals and prescriptions are available as clinical codes (eg, Read codes).

### Study design and population

This retrospective cohort study included patients (≥18 years) who had no previous IBD diagnosis and who had their first FC test between 1 January 2006 and 31 December 2016. Eligible patients had also contributed at least 12 months of data to THIN before their study start

date. FC tests were identified using all six Read codes available to GPs to record an FC test.

### Classification of FC test results

FC tests with a numeric result of >50 and >100 µg/g were considered positive in two separate analyses. We included a post hoc analysis at the threshold of 250 µg/g in light of the latest recommendations of a pathway using two cut-offs.[10 14] FC tests without results, with invalid results, qualitative results only, missing units of measurement or units of measurement other than µg/g were excluded from the analysis.

### Definition of disease

IBD was defined as a record of a Read code for ulcerative colitis, Crohn's disease, indeterminate colitis or microscopic colitis. Read code lists were adapted from those used in previous literature.[15–17] This was supplemented with the record of an IBD-specific prescription to increase the sensitivity of the algorithm. IBS and CRC were defined by any clinical code that suggested a definitive diagnosis. Abdominal symptoms were not considered sufficiently specific for an IBS diagnosis. We excluded benign and precancerous stages of CRC. In three different analyses a diagnosis was considered when it was recorded within 6, 12 or 24 months of the FC test date to explore the impact of late diagnoses in patients with sufficient follow-up after FC testing. We considered the following symptoms to assess eligibility for testing: diarrhoea, abdominal pain, constipation, bloating and change in bowel habit. See online supplemental table 1 for the final code lists of these variables.

### Definition of referral

Our definition of IBD was based on the assumption that IBD diagnoses were only recorded following confirmatory testing in secondary care. We explored this assumption by investigating FC testing in the referred population only. We considered a referral to any specialty that was made within 6 weeks of the index FC test to be a referral in response to FC testing.

### Statistical analysis

Data management and analysis were undertaken in R V.3.6.1 (Vienna, Austria).[18] The main outcomes of the analysis were sensitivity, specificity, and positive (PPV) and negative predictive values (NPV). These were calculated in two ways: (1) Patients with a recorded IBD diagnosis were classed as disease positive and those without an IBD diagnosis (IBS, CRC, alternative diagnosis, no diagnosis) were classed as disease negative. (2) Patients with an IBD diagnosis were classed as disease positive and only those with an IBS diagnosis were classed as disease negative. CIs for proportions were computed using the Wilson method.[19] We used the Koopman method to compute CIs for likelihood ratios.[20]

We used receiver operating characteristic (ROC) curves to display the accuracy of the FC test at test positive thresholds between 33 and 300 µg/g. This allowed the

inclusion of results recorded as >300 and <33 µg/g. ROC curves for different follow-up times were compared using Venkatraman's method for two unpaired ROC curves in the absence of a feasible paired comparison because data were of different sizes.[21]

We computed proportions of patients referred with a true positive (TP), false positive (FP), false negative (FN) and true negative (TN) FC test result in order to investigate the effect the FC test results have on patient management. If the test results had no impact on referral, the proportions across groups were expected to be similar. If GPs act on the test result (refer positives and manage negatives), the proportion of patients referred with TP and FP results was expected to be (1) close to one, (2) equal and (3) greater than those for FN and TN results (both close to 0). The proportions of referred patients were compared using the two proportions z-test.

Sensitivity analyses were conducted in order to investigate the effects different assumptions have on the test accuracy estimates. Specifically, we investigated including all units of measurements; including qualitative results in addition to quantitative results; defining IBD by Read code only; and confining analyses to only those patients referred to secondary care. We also explored the effects of excluding patients not eligible for the FC testing as according to national guidance (patients with an age >50 years or symptoms for <6 weeks).

### Patient involvement

A patient advisory group was actively involved in the funding application for the project. The group provided a patient perspective in the design of the study and the interpretation of study findings.

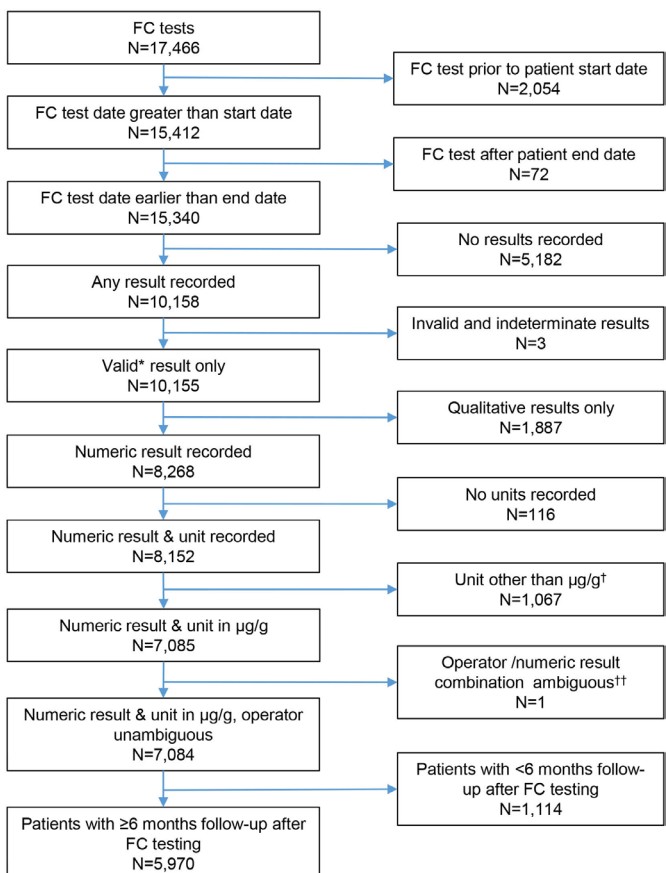

**Figure 1** Flow diagram of inclusion criteria of faecal calprotectin (FC) tested patients into test accuracy study. *Results not coded invalid or indeterminate in THIN. †Fifteen different units of measurements were used to record FC levels. µg/g is the most commonly used in publications and laboratories. ††FC level of >30µg/g ambiguous at 50µg/g and 100µg/g thresholds

### RESULTS

A total of 17 466 first time FC tests in a total of 6 965 853 adult primary care patients were recorded in the study period, and 5970 were included in the analysis (figure 1). Of these, 1987 (32%) had an IBS diagnosis, 208 (3.5%) an IBD diagnosis, 31 (0.5%) a CRC diagnosis, 78 (1%) had an IBD and an IBS diagnosis, 2 had an IBD and a CRC diagnosis, and 3754 (63%) had no diagnosis recorded at any time after the FC test (see online supplemental table 2). Of the 5970 patients, 1434 (24%) had an IBS diagnosis recorded at the time of FC testing. The proportion of patients with an IBD or IBS diagnosis was slightly higher in the included group than in the excluded group (see online supplemental table 2).

### Test accuracy of FC testing for IBD

In 5970 patients with at least 6 months of follow-up available after the FC test date (figure 1), the sensitivity of the FC test in detecting IBD at a threshold of 50 µg/g was 92.9% (95% CI 88.6% to 95.6%). Specificity was 61.5% (95% CI 60.2% to 62.7%). The positive predictive value (PPV) was low at 8.1% (95% CI 7.1% to 9.2%), while the NPV was high (99.6% (95% CI 99.3% to 99.7%)) at an

IBD prevalence of 3.5% (table 1). The positive likelihood ratio was 2.41 (95% CI 2.28 to 2.52), which suggests that FC testing only slightly increases the probability of IBD. The negative likelihood ratio was 0.12 (95% CI 0.07 to 0.19), which suggests strong evidence to rule out IBD. The inclusion of IBD cases that were recorded up to 12 (analysis of 4793 FC tested patients) and 24 months (2662 patients) following an FC test had no significant effect on the measures of test accuracy (table 1). Small changes observed in the point estimates may be due to the different population size and the different make-up of the study population. Eighty-seven per cent (2096/2414) of patients with a positive FC test result had no recorded diagnosis of IBD, IBS or CRC 6 months after the FC test date. Of these, 501 had a previous diagnosis of IBS or CRC. At the end of the study period, there were still 1470 (61%) patients with a positive test but without a diagnosis of interest. These patients could either have had a missed diagnosis, no diagnosis or an alternative diagnosis, which was not included in our study. For patients with a negative FC test, 92% (3277/3556) had none of the three

**Table 1** Test accuracy measures of faecal calprotectin (FC) testing at a threshold of 50 µg/g

| Time* | N | TP | FP | FN | TN | Sensitivity, % (95% CI) | Specificity, % (95% CI) | PPV, % (95% CI) | NPV, % (95% CI) |
|---|---|---|---|---|---|---|---|---|---|
| 6 months | 5970 | 195 | 2219 | 15 | 3541 | 92.9 (88.6 to 95.6) | 61.5 (60.2 to 62.7) | 8.1 (7.1 to 9.2) | 99.6 (99.3 to 99.7) |
| 12 months | 4793 | 198 | 1790 | 16 | 2789 | 92.5 (88.2 to 95.3) | 60.9 (59.5 to 62.3) | 10 (8.7 to 11.4) | 99.4 (99.1 to 99.6) |
| 24 months | 2662 | 129 | 1067 | 11 | 1455 | 92.1 (86.5 to 95.6) | 57.7 (55.8 to 59.6) | 10.8 (9.2 to 12.7) | 99.2 (98.7 to 99.6) |

*Time following FC testing to consider a record of inflammatory bowel disease.
FN, false negatives; FP, false positives; N, sample size; NPV, negative predictive value; PPV, positive predictive value; TN, true negatives; TP, true positives.

diagnoses recorded at 6 months and 64% (2284/3556) had none of the three diagnoses at any time after FC testing.

The ROC curves in figure 2 describe the sensitivity and false positive rate (1-specificity) for the three different follow-up times at additional thresholds. The 50 µg/g threshold prioritises sensitivity over specificity in all three analyses. There was generally good agreement between the three curves (E=0.0016758, p=0.406 comparing 6 months with 24 months of follow-up).

### Increasing the FC threshold

Figure 3 shows that an increase in the threshold from 50 to 100 µg/g affects the specificity more than the sensitivity (online supplemental table 3). Increasing the threshold reduces the number of false positives by 875 at the expense of missing an additional 14 cases of IBD in a population of 5970 tested patients, suggesting that the proportion of IBD cases missed due to this threshold difference is less than 7% (14/210). IBD cases missed were 31% (65/210) when we changed the threshold to 250 µg/g (online supplemental table 3).

### Restricting the population to patients with IBD or IBS

Changing the definition of the non-diseased group from non-IBD to IBS reduced the number of false positives and true negatives. Consequently, the point estimates of specificity and PPV increased by about 10% and 60%, respectively, at the 50 µg/g threshold (table 2). However, the dataset was small (576 vs 5970) and, thus, increases the potential for sampling error. This may limit the clinical applicability of the findings.

### Sensitivity analyses

Overall, the sensitivity analyses revealed greater uncertainty in the estimate of specificity than sensitivity (table 3). Inclusion of numerical results reported in other units of measurement had no impact on estimates of test accuracy. The majority of qualitative results were positive at the 50 µg/g threshold, which may indicate preferential recording of positive test results using qualitative codes. In this sensitivity analysis, the specificity was lower due to the great number of false positives. Only 569 patients tested had symptoms recorded that met the national guidance eligibility criteria. The recording of symptoms was not associated with the FC test result. Test accuracy measures in this subset did not differ from those estimated using the full dataset. Reclassifying 79 patients as IBD negative

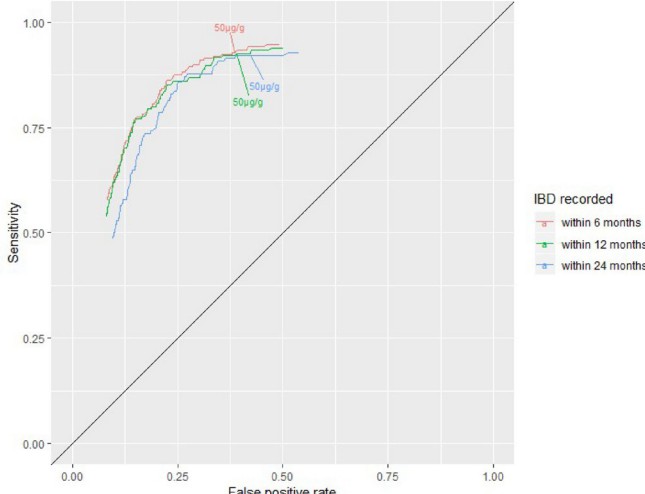

**Figure 2** Receiver operating characteristic curve of sensitivity and false positive rate (1-specificity) for thresholds of 33–300 µg/g for the clinical question inflammatory bowel disease (IBD) versus non-IBD for tests with at least 6 months (n=5970), 12 months (n=4793) and 24 months (n=2662) follow-up available after the faecal calprotectin test date.

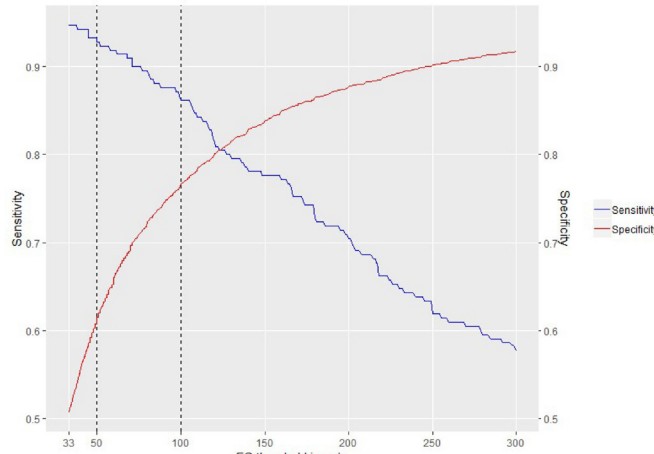

**Figure 3** Sensitivity and specificity at thresholds of 33–300 µg/g for inflammatory bowel disease (IBD) versus non-IBD and IBD recorded within 6 months following the faecal calprotectin (FC) test date (n=5970).

**Table 2** Test accuracy measures for faecal calprotectin (FC) testing at the threshold of 50 µg/g for the detection of IBD recorded within 6 months of FC testing considering two different definitions of the non-diseased group

| | N | TP | FP | FN | TN | Sensitivity, % (95% CI) | Specificity, % (95% CI) | PPV, % (95% CI) | NPV, % (95% CI) |
|---|---|---|---|---|---|---|---|---|---|
| IBD vs non-IBD | 5970 | 195 | 2219 | 15 | 3541 | 92.9 (88.6 to 95.6) | 61.5 (60.2 to 62.7) | 8.1 (7.1 to 9.2) | 99.6 (99.3 to 99.7) |
| IBD vs IBS | 576 | 195 | 104 | 15 | 262 | 92.9 (88.6 to 95.6) | 71.6 (66.8 to 76) | 65.2 (59.7 to 70.4) | 94.6 (91.3 to 96.7) |

FN, false negative; FP, false positive; IBD, inflammatory bowel disease; IBS, irritable bowel syndrome; N, sample size; NPV, negative predictive value; PPV, positive predictive value; TN, true negative; TP, true positive.

who had an IBD diagnosis affirmed by an IBD-specific drug code only (Read code negative) had no impact on the test accuracy estimates. Finally, of the 5970 FC tests, 52% resulted in a referral. The specificity of FC testing in this subgroup was lower because of the non-referral of mainly true negatives.

### Impact of test results on referral decisions

Table 4 shows the classification of all included patients and those referred within 6 weeks of their FC test according to their test result and disease status. If GPs referred patients based on positive FC test results, they would have referred 2414 out of 5970 (40.4%) included patients. This would have detected 195 out of 210 (92.9%) IBD cases. However, 6 weeks after FC testing, 3078 out of 5970 (51.6%) patients were referred, including 1610 out of 3556 (45.3%) FC negative patients and only 149 out of 210 (71.0%) IBD cases. Therefore, the test result was not the single determining factor for or against a timely referral. Patients with true positive and false negative FC test results were equally likely to be referred for further tests (70.8% vs 73.3%, z=2, p=0.8371). Both groups were referred more frequently than patients with false positive

(59.9%) (z=3, p=0.0028) or true negative results (45.3%) (z=7, p<0.0001). This may suggest that GPs used alternative cues (in the history, examination or other tests) which raised their suspicion of IBD. Overall, GP assessment plus FC testing may be worse than FC testing alone. Compared with FC testing alone, the number of patients with a true positive result was lower for GP assessment plus testing because GPs did not refer all test positive patients within 6 weeks. GPs referred some patients with a false negative FC test result on the test, but not enough to make up for not referring patients with true positive results on the test.

## DISCUSSION
### Study findings

After excluding FC tests with no result or an ambiguous result, the sensitivity of the FC test in detecting IBD at the 50 µg/g threshold was high (92.9% (95% CI 88.6% to 95.6%)) but the specificity was modest (61.5% (95% 60.2% to 62.7%)). Moreover, the sensitivity analyses demonstrated greater uncertainty in our estimates of the specificity. The high false positive rate may be irrelevant

**Table 3** Sensitivity analyses compared with primary analysis of FC testing for IBD vs non-IBD at a threshold of 50 µg/g with IBD recorded within 6 months

| Analysis | N | TP | FP | FN | TN | Sensitivity,% (95% CI) | Specificity, % (95% CI) | PPV, % (95% CI) | NPV, % (95% CI) |
|---|---|---|---|---|---|---|---|---|---|
| Primary analysis | 5970 | 195 | 2219 | 15 | 3541 | 92.9 (88.6 to 95.6) | 61.5 (60.2 to 62.7) | 8.1 (7.1 to 9.2) | 99.6 (99.3 to 99.7) |
| FC tests with any unit | 6758 | 220 | 2481 | 17 | 4040 | 92.8 (88.8 to 95.5) | 62 (60.8 to 63.1) | 8.1 (7.2 to 9.2) | 99.6 (99.3 to 99.7) |
| Qualitative results included* | 7000 | 261 | 3177 | 15 | 3547 | 94.6 (91.2 to 96.7) | 52.8 (51.6 to 53.9) | 7.6 (6.8 to 8.5) | 99.6 (99.3 to 99.7) |
| Eligible according to national guidelines† | 569 | 20 | 194 | 1 | 354 | 95.2 (77.3 to 99.2) | 64.6 (60.5 to 68.5) | 9.3 (6.1 to 14) | 99.7 (98.4 to 100) |
| IBD defined by Read code only | 5970 | 148 | 2266 | 6 | 3550 | 96.1 (91.8 to 98.2) | 61 (59.8 to 62.3) | 6.1 (5.2 to 7.2) | 99.8 (99.6 to 99.9) |
| Referred only‡ (within 6 weeks) | 3078 | 138 | 1330 | 11 | 1599 | 92.6 (87.3 to 95.8) | 54.6 (52.8 to 56.4) | 9.4 (8 to 11) | 99.3 (98.8 to 99.6) |

*With a record of an upper reference range of 50 or above.
†Patients <50 years and abdominal symptoms for >6 weeks.
‡Assumption that referred patients have high probability of disease verification by colonoscopy.
FC, faecal calprotectin; FN, false negatives; FP, false positives; IBD, inflammatory bowel disease; N, sample size; NPV, negative predictive value; PPV, positive predictive value; TN, true negatives; TP, true positives.

**Table 4** Proportion of patients referred by test result and disease status

| | TP | FP | FN | TN | Total |
|---|---|---|---|---|---|
| Classification of included patients in primary analysis based on FC test result, n | 195 | 2219 | 15 | 3541 | 5970 |
| Classification of FC tested patients with referral based on FC test result, n | 138 | 1330 | 11 | 1599 | 3078 |
| Proportion of patients in primary analysis referred, % | 70.8 | 59.9 | 73.3 | 45.3 | 51.6 |

FC, faecal calprotectin; FN, false negative; FP, false positive; TN, true negative; TP, true positive.

in clinical practice if the group represents alternative inflammatory conditions that warrant further assessment. However, a broader disease category in primary care including other organic conditions in addition to IBD resulted in sensitivity estimates ranging from 64% to 94%.[22–24] As the conditions may follow different pathways, interpretation of a test positive result would still be challenging. Restricting the group without IBD to those with a confirmed IBS record increased the specificity by 11 percentage points but has little meaning for clinical practice because it is an artificial patient population. Extending the follow-up time to allow for late recordings of IBD had no impact on sensitivity and specificity. The ROC analysis revealed that the recommended FC threshold of 50 µg/g prioritises sensitivity over specificity.

Changing the threshold from 50 to 100 µg/g had a greater effect on the specificity than the sensitivity. For clinical practice, this could translate into fewer referrals of non-IBD patients to gastroenterology at the expense of missing an additional nearly 7% of IBD cases. This interpretation is purely based on the logic of test results. Investigation of the proportion of referrals of patients with true positive, false positive, false negative and true negative test results showed that the actual management of tested patients may not clearly follow the expected behaviour of referring positives and managing test negative patients.

This interpretation assumes that referrals were recorded consistently for those with and without an IBD record and irrespective of FC test outcome. We interpret the findings with caution because we could not confirm the final classification of more than half of the tested patients and missing IBD cases cannot be excluded. Furthermore, all patients with IBD would have been eventually referred according to the definition of an IBD diagnosis, which assumed that IBD diagnoses are only recorded following confirmation by a gastroenterologist and invasive testing. Further, 61 out of 210 (29.0%) IBD cases were either referred after 6 weeks or their referral was not recorded.

## Strengths and limitations

One benefit of using routine data to estimate a test's accuracy is that the sample size is likely to be larger than the usual cross-sectional studies used to evaluate a test. This is particularly useful for target disorders such as IBD which have a low prevalence.[25] We applied strict eligibility criteria to ensure the reliability of reported test accuracy measures and we explored the effects of different follow-up times on the definition of IBD. In the case of FC testing in routine primary care, where it is not feasible to expose all tested patients to colonoscopy for disease verification, we were able to provide an answer to the question of the accuracy of the FC test as it is used in UK general practice. Our study shows that FC testing together with the GP's decision-making is an effective tool to exclude IBD in test negative patients.

The study may have suffered from selection bias because the inclusion criterion was a recorded FC test rather than symptomatic patients.[26] However, we believe this to be minimal when considering the point estimates and CIs of test accuracy measures in the sensitivity analysis of patients with eligible symptoms.

The exclusion of close to half of the FC tests without a numerical test result raises concerns over the generalisability of the findings because excluded tests were statistically different from the included tests in some variables tested. However, the tests were highly powered because of the large dataset and the clinical significance of the differences is highly uncertain.

Although imputation methods are often used to mitigate against missing data, these were not implemented for three reasons. First, in observational data, observed variables are not sufficient to account for the differences between missing and observed values.[27] Second, for imputation methods to be meaningful, the proportion of missing data should be small and the missing subpopulation should be sufficiently similar to the wider population neither of which could be confirmed. Third, in order to build reliable imputation models, more meaningful variables are needed including severity of symptoms, assay type, referral to gastroenterology and information on disease verification method, none of which is available in the THIN database. Therefore, potential biases could only be addressed by sensitivity analyses.[27]

The proportion of patients without a diagnosis following testing was large and could be due to any of the following: missed IBD, unrecorded IBS, other conditions not considered as study variables, or diagnoses recorded as free text and thus not available to the study. Also the research literature was too heterogeneous to be able to estimate the prevalence of other conditions accurately. For example, across five studies,[28–32] the number of conditions varied widely and the prevalence ranged from 4% (16/399)[28] to 61% (422/694).[31] We were unable to predict IBD using other variables, such as symptoms and comorbidities, in the non-IBD group due to the lack of strong predictors of IBD among the study variables. Without any further

information on the characteristics of the non-IBD cases, their true disease status remains uncertain.

Colonoscopy with histology is the preferred reference standard for diagnosis of IBD. However, information on secondary care events is not fully captured in THIN. In the absence of sufficient information on colonoscopies, we used a combination of Read and drug codes as a reference standard under the assumption that IBD is not generally coded without confirmatory colonoscopy. Our pragmatic study for test accuracy aimed to establish the test performance of FC testing as it is used in clinical practice. The ideal reference standard is generally not feasible in these studies. While this does not invalidate the study, the study aim and the performance of the reference standard need to be considered in the interpretation of the study findings.

We were interested in the broad category of IBD. While ulcerative colitis and Crohn's disease are the two predominant diagnoses included as part of IBD, they are considered as the two extremes of a spectrum of chronic gut disorders.[33–35] IBD cannot be classified in 20%–30% of patients at first presentation, resulting in IBD unclassified cases.[33 34] We also included unclassified forms of IBD such as indeterminate colitis and microscopic colitis to present the complete picture of IBD in primary care.

Our results are broadly in line with pooled estimates from a meta-analysis of 14 secondary and primary are studies,[11] which may suggest that the pooled test accuracy measures are applicable to the FC tested population in primary care. At the thresholds investigated, our sensitivity and specificity estimates are within the range of estimates from published primary care studies.[28 36 37] However, previous primary care studies included small numbers of cases, while our study provides narrower CIs around test accuracy estimates. Crucially, our study shows that the specificity of 94% (95% CI 73% to 99%) reported in NICE guidance is not applicable to the primary care setting,[8] where the broader definition of the non-IBD group means that the specificity and the PPV of FC testing is much lower. Therefore, the interpretation of a test positive result may be challenging. Furthermore, the higher number of patients with a false positive result may not achieve the reduction of referrals to colonoscopy as anticipated by NICE guidance.

### Implications for clinical practice

This study indicates that FC testing, when used by GPs, does not miss many cases of IBD in a low prevalence setting (NPV 99.6% (95% CI 99.3% to 99.7%)). GPs could, therefore, use the test confidently in ruling out IBD if the test is negative. However, the test does not perform as well in identifying IBD as modelled by the national guidance. The PPV of the FC test in identifying IBD is low, as essentially FC is a marker of intestinal inflammation rather than a specific marker for IBD.

The data also suggested that the higher threshold of $100\,\mu g/g$ may be appropriate for routine primary care. This is under the understanding that the test is not used

in isolation and the objective is to reduce the number of unnecessary colonoscopies. Patients with missed IBD would be assumed to represent and be reconsidered for referral within an appropriate time frame. However, the latest FC pathway recommends two cut-offs (100 and $250\,\mu g/g$)[10 14] with an intermediate test range for retesting. The additional IBD cases missed at initial testing with this strategy may be inappropriately high. In this study, we have not considered the feasibility of this latest pathway but it is likely to be more difficult to interpret for GPs.

Our study identified a number of inconsistencies between the intended use by the national guidelines and the actual use by GPs. First, GPs tested an unexpectedly high proportion of patients with IBS, which may suggest that GPs use the test to reassure patients with IBS. Second, practitioners referred proportionately more patients with false negative results than false positive and true negative results. This suggests that practitioners use other cues for referral rather than test results alone. Third, close to half of all negative FC tested patients were referred, which suggests that GPs and/or patients may have limited confidence in the test. The current guidelines may not be applicable to general practice and the modelled cost-effectiveness of FC testing in general practice may be overoptimistic. When developing national guidance, models should more closely represent general practice, rather than simply extrapolating from secondary care. This study shows for the first time the benefits and shortcomings of the performance of FC testing in a large dataset of primary care patients. It should help improve GPs' confidence in their diagnostic ability in this area and in test use and interpretation, especially where there is a negative test.

**Correction notice** This article has been corrected since it first published. In this article the word "IMRD-UK" has been replaced by "THIN" in the title and throughout the article because the database did not change its name with the change in ownership of the database.

**Acknowledgements** We would like to thank the patient advisory group of the project 'What is the role of faecal calprotectin testing in primary care' for their input.

**Contributors** KF, ST-P, BHW, RR and AC designed the study. RR extracted the data from THIN and created the datasets. KF, ST-P and BHW carried out the analysis. KF, BHW, ST-P and AC contributed to the interpretation of the findings. KF drafted the manuscript. All authors critically revised the manuscript and approved the final version. KF takes responsibility for the integrity and accuracy of the data analysis. KF acts as guarantor. The corresponding author attests that all listed authors meet authorship criteria and that no others meeting the criteria have been omitted.

**Funding** KF is funded by a National Institute for Health Research (NIHR) DRF award (DRF-2016-09-038) for this research project. AC is supported by the National Institute for Health Research (NIHR) Applied Research Collaboration (ARC) West Midlands. This report presents independent research funded by the National Institute for Health Research (NIHR). The views expressed are those of the author and not necessarily those of the NHS, the NIHR or the Department of Health and Social Care. The funder had no role in the study design, data collection, data analysis and interpretation, writing of the report or the decision to submit for publication.

**Competing interests** All authors have completed the Unified Competing Interest form (available on request from the corresponding author) and declare the following: KF is funded by the NIHR through a doctoral research fellowship. ST-P reports grants from the NIHR fellowship for Karoline Freemen. AC is supported

by the NIHR ARC West Midlands initiative. BHW received grants from the Medical Research Council. RR has no conflicts to declare.

**Patient consent for publication**  Not required.

**Ethics approval**  The study was approved by the Independent Scientific Advisory Committee (protocol number 17THIN089).

**Provenance and peer review**  Not commissioned; externally peer reviewed.

**Data availability statement**  No data are available. The THIN dataset cannot be shared under the data sharing agreement with the University of Birmingham on behalf of IQVIA.

**ORCID iDs**
Karoline Freeman http://orcid.org/0000-0002-9963-2918
Sian Taylor-Phillips http://orcid.org/0000-0002-1841-4346

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
