## [Reviewer comments · BMJ Open]

ARTICLE DETAILS

TITLE (PROVISIONAL)	Test Accuracy of Faecal Calprotectin for Inflammatory Bowel Disease in UK Primary Care: Retrospective Cohort Study of the IMRD-UK data
AUTHORS	Freeman, Karoline; Taylor-Phillips, Sian; Willis, Brian; Ryan, Ronan; Clarke, Aileen

VERSION 1 – REVIEW

REVIEWER	Joaquín Cubiella Department of Gastroenterology Complejo Hospitalario Universitario de Ourense Ourense, Spain.
REVIEW RETURNED	19-Sep-2020

GENERAL COMMENTS	I have read with interest the manuscript entitled “Test Accuracy of Faecal Calprotectin for Inflammatory Bowel Disease in UK Primary Care: Retrospective Cohort Study of the IMRD-UK data”. It tries to address a difficult issue: what is the effect of the non-invasive available biomarkers for the diagnosis and referral of patients with significant bowel disease? Major comment: My main concern is derived from the IBD diagnosis criteria. What is the accuracy of the gold standard used for IBD diagnosis in comparison with the endoscopic diagnosis?. In fact, the authors do not include which are the IBD specific prescription. They have to take into account that certain prescriptions are not specific of IBD: azathioprine, infliximab. I am worried also of the criteria used for the IBS diagnosis. Some of the codes included to detect IBD are not part of the diagnosis of IBD, but other indeterminate colitis: • J436.00 Microscopic colitis• J436000 Collagenous colitis• J436100 Lymphocytic colitis• J438.00 Left sided colitis• J4z3.00 Non-infective colitis NOS• J4z4.00 Non-infective sigmoiditis NOS• J4z6.00 Indeterminate colitis Minor comments: • Abstract: Authors must include the threshold used to determine the calprotectin accuracy (50 or 100)• Numbers of the final diagnosis and the diagnostic accuracy at each period do not fit: $208+78+2= 288$ patients had a final diagnosis of IBD. In contrast, at 6, 12 and 24 months, 210, 214 and 140 patients had a IBD diagnosis. In fact, I would suggest to delete the analysis at the 24 month period, as long as only 44% of the initial patient are evaluated and the follow-up is too long.
--

	 • Authors should include in the statistical analysis the ROC curve analysis and the likelihood ratios. On account the ratio post-pre PPV 8.1%/4.5% for IBD detection, the + LR should be below 2, thus with an extremely low discriminative capacity. • Analysis restricted to differentiate IBD from IBS. This analysis has little value, as long as GPs will have to evaluate patients with abdominal symptoms and the IBS will be based on clinical persistent symptoms with no organic origin. In fact, it would be relevant if authors produce an analysis of the diagnostic accuracy of the calprotectin in patients with persistent symptoms. In this sense, it would be extremely relevant to know the PPV in relation with the symptoms persistence that produced the calprotectin determination • In the impact of test results on referral decisions, the authors are including an interpretation of the results. They should include any interpretation in the discussion section. • Figure 2: It should be clearly improved, and the AUC added
--	--

REVIEWER	BOUZIANE Amal Mohammed V University in Rabat Morocco
REVIEW RETURNED	26-Sep-2020

GENERAL COMMENTS	General remark: The study tries to estimate the test accuracy of faecal calprotectin (FC) for inflammatory bowel disease (IBD) in the primary care setting. There are some consistent evaluated data. However, Likelihood ratios and their interpretation are not described. These have several properties that make them more useful clinically than other measures of diagnostic test performance. Sensitivity and specificity are calculated for the test when the disease statuses of the patients are known. Positive and negative predictive values are affected by changes in the prevalence of disease. Therefore, the interpretation of a test is highly dependent on the context in which it is used. Likelihood ratios do not have the drawbacks of mentioned measures of test performance. It would be essential, to calculate and interpret positive and negative Likelihood ratios. Specific remarks: • Abstract: IBS should be explained for the first time. • Introduction: Given the aim of the study, it is important to indicate the prevalence of inflammatory bowel disease (IBD) in UK and in the world.
--

VERSION 1 – AUTHOR RESPONSE

Reviewer's comments	Authors' responses
Reviewer 1	
My main concern is derived from the IBD diagnosis criteria. What is the accuracy of the gold standard used for IBD diagnosis in comparison with the endoscopic diagnosis?	We completely agree that the reference standard is a limitation of test accuracy studies using routine data as we do not know the accuracy of lists of drugs and codes. We have discussed this limitation. However, using a

	coded IBD diagnosis as confirmation of disease is based on the knowledge that IBD diagnoses are not generally coded without confirmatory testing. In clinical practice the diagnosis of IBD, in general, will be after a colonoscopy. We recognise this bias and concluded that our pragmatic test accuracy study reflects how FC testing works in real life rather than proving the accuracy of FC testing in categorising IBD and IBS patients, which has been done before. We added the following paragraph to strengthen the discussion around verification using routine data: Colonoscopy with histology is the preferred reference standard for diagnosis of IBD. However, information on secondary care events is not fully captured in the IMRD. In the absence of sufficient information on colonoscopies, we used a combination of Read and drug codes as a reference standard under the assumption that IBD is not generally coded without confirmatory colonoscopy. Our pragmatic study for test accuracy aimed to establish the test performance of FC testing as it is used in clinical practice. The ideal reference standard is generally not feasible in these studies. While this does not invalidate the study, the study aim and the performance of the reference standard needs to be considered in the interpretation of the study findings.
In fact, the authors do not include which are the IBD specific prescription. They have to take into account that certain prescriptions are not specific of IBD: azathioprine, infliximab.	We only included IBD specific prescriptions in the definition of IBD. Azathioprine and infliximab were not included. We have included the drug code list in the appendix for clarification.
I am worried also of the criteria used for the IBS diagnosis.	IBS coding might be less reliable than IBD coding since there is no diagnostic marker at present to confirm IBS, which is a diagnosis by exclusion (i.e. by ruling out other conditions). Therefore, there may be more IBS codes missing than IBD codes. However, the non-diseased group in our study is non-IBD rather than IBS (which is not applicable to primary

	care), so missing codes are still correctly classified as non-IBD.
Some of the codes included to detect IBD are not part of the diagnosis of IBD, but other indeterminate colitis:  • J436.00 Microscopic colitis • J436000 Collagenous colitis • J436100 Lymphocytic colitis • J438.00 Left sided colitis • J4z3.00 Non-infective colitis NOS • J4z4.00 Non-infective sigmoiditis NOS • J4z6.00 Indeterminate colitis 	IBD is a heterogeneous group of disorders. Crohn's disease and ulcerative colitis are considered as the two extremes of a spectrum of chronic gut disorders. IBD cannot be classified in 20% to 30% of patients at first presentation resulting in IBD unclassified (indeterminate colitis) cases. [Goebes 2003] We therefore class these codes as IBD which are different forms of colitis which literally means inflammation of the bowel. The literature shows that there is considerable variation in the definition of IBD for instance in epidemiological studies of IBD. We agree that if IBD is strictly defined as UC and CD, indeterminate colitis would not be included in the definition of IBD. However, this distinction can only be made in specialist care and it is not useful for general practice. FC testing is an inflammatory marker and is thought to aid the decision making process whether to refer patients with abdominal symptoms to specialist care for further investigation. This includes patients with indeterminate colitis who are expected to follow the same patient pathway. Geboes K, De Hertogh G. Indeterminate colitis. Inflamm Bowel Dis. 2003;9(5):324-31. doi: 10.1097/00054725-200309000-00007 We have added the following paragraph into the discussion to address the reviewer's point. We were interested in the broad category of inflammatory bowel disease. Whilst ulcerative colitis and Crohn's disease are the two predominant diagnoses included as part of IBD they are considered as the two extremes of a spectrum of chronic gut disorders.³³⁻³⁵ IBD cannot be classified in 20% to 30% of patients at first presentation resulting in IBD unclassified cases.^{33, 34} We also included unclassified forms of IBD such as indeterminate colitis and

	microscopic colitis to present the complete picture of IBD in primary care.
Abstract: Authors must include the threshold used to determine the calprotectin accuracy (50 or 100)	The threshold relating to the results is clearly reported in the abstract. Sensitivity, specificity, positive and negative predictive values for the differential of IBD versus non-IBD and IBD versus irritable bowel syndrome (IBS) at the 50µg/g and 100µg/g thresholds. Sensitivity, specificity, positive and negative predictive values were 92.9% (88.6% to 95.6%), 61.5% (60.2% to 62.7%), 8.1% (7.1% to 9.2%) and 99.6% (99.3% to 99.7%), respectively at the threshold of 50µg/g. Raising the threshold to 100µg/g missed less than 7% additional IBD cases.
Numbers of the final diagnosis and the diagnostic accuracy at each period do not fit: 208+78+2= 288 patients had a final diagnosis of IBD. In contrast, at 6, 12 and 24 months, 210, 214 and 140 patients had a IBD diagnosis.	We apologise that the reporting was not sufficiently clear. The study first reports diagnoses recorded at any time after FC testing to characterise the study population. The analyses of test accuracy study only considered IBD diagnoses within 6 months, 12 months and 24 months, respectively as reported in the methods. We have changed the following sentence to clarify what we report in the characterisation of the FC tested population: 1,987 (32%) had an IBS diagnosis, 208 (3.5%) an IBD diagnosis, 31 (0.5%) a colorectal cancer (CRC) diagnosis, 78 (1%) had an IBD and an IBS diagnosis, 2 had an IBD and a CRC diagnosis and 3,754 (63%) had no diagnosis recorded at any time after the FC test (see online Supplementary Table 2).
In fact, I would suggest to delete the analysis at the 24 month period, as long as only 44% of the initial patient are evaluated and the follow-up is too long.	The three different follow-up times were included to explore the effect of late diagnosis on test accuracy. We have shown that late diagnosis has little impact on test accuracy which supports our study findings. We have included the 24 months follow-up based on 1) the study by Varicka et al. 2012 who report a IQR of 3-24 months from symptoms onset to CD diagnosis and 2) patient accounts from our PPI

	group who reported long delays in diagnosis (“a hellish 10 or 18 months”). We prefer to include the analysis for completion. Vavricka SR, Spigaglia SM, Rogler G, Pittet V, Michetti P, Felley C, Mottet C, Braegger CP, Rogler D, Straumann A, Bauerfeind P, Fried M, Schoepfer AM. Systematic evaluation of risk factors for diagnostic delay in inflammatory bowel disease. Inflamm Bowel Dis. 2012;18(3):496-505. doi: 10.1002/ibd.21719.
Authors should include in the statistical analysis the ROC curve analysis and the likelihood ratios. On account the ratio post-pre PPV 8.1%/4.5% for IBD detection, the + LR should be below 2, thus with an extremely low discriminative capacity.	We have added the LR+ to the analysis for 6 months (PPV 8.1% and prevalence 3.5%) and report the following paragraph. The positive likelihood ratio was 2.41 (95% CI 2.28 to 2.52) which suggests that FC testing only slightly increases the probability of IBD disease. The negative likelihood ratio was 0.12 (95% CI 0.07 to 0.19) which suggests strong evidence to rule out IBD. The LR+ 2.41 and PPV 8.1% refer to results at 6 months. The prevalence at 6 months was 3.5% (see table 1 in the manuscript). The prevalence of 4.5% is for the analysis at 12 months. We prefer to keep to our initial decision not to report the AUCs as they don't add anything to the interpretation of the test accuracy of FC testing in primary care. Only the test accuracy of FC testing at specific clinically relevant threshold is useful for clinicians and their decision making.
Analysis restricted to differentiate IBD from IBS. This analysis has little value, as long as GPs will have to evaluate patients with abdominal symptoms and the IBS will be based on clinical persistent symptoms with no organic origin.	We completely agree. It was important for us to report the impact of an analysis of IBD versus IBS on the test accuracy because UK national guidance was based on estimates from such analysis and they recommend FC testing for the differential of IBD versus IBS.
In fact, it would be relevant if authors produce an analysis of the diagnostic accuracy of the calprotectin in patients with persistent symptoms. In this sense, it would be extremely	We agree that a test accuracy study of eligible patients based on persistent abdominal symptoms would be useful. We included a sensitivity analysis of patient with eligible

relevant to know the PPV in relation with the symptoms persistence that produced the calprotectin determination.	symptoms (see Table 3). The different size in study populations (569 vs 5970 patients) revealed little overlap between tested and eligible patients. Furthermore, identifying patients with persistent symptoms is not feasible as this cannot be easily operationalised using routine electronic health care records of coded clinical data.
In the impact of test results on referral decisions, the authors are including an interpretation of the results. They should include any interpretation in the discussion section.	Thank you, we have moved the relevant paragraph below into the discussion. This interpretation assumes that referrals were recorded consistently for those with and without an IBD record and irrespective of FC test outcome. We interpret the findings with caution because we could not confirm the final classification of more than half of the tested patients and missing IBD cases cannot be excluded. Furthermore, all IBD patients would have been eventually referred according to the definition of an IBD diagnosis, which assumed that IBD diagnoses are only recorded following confirmation by a gastroenterologist and invasive testing. 61/210 (29.0%) IBD cases were either referred after 6 weeks or their referral was not recorded.
Figure 2: It should be clearly improved, and the AUC added	Please see our response about AUCs above.
Reviewer 2	
Sensitivity and specificity are calculated for the test when the disease statuses of the patients are known. Positive and negative predictive values are affected by changes in the prevalence of disease. Therefore, the interpretation of a test is highly dependent on the context in which it is used. Likelihood ratios do not have the drawbacks of mentioned measures of test performance. It would be essential, to calculate and interpret positive and negative Likelihood ratios.	We have added the following paragraph reporting the LRs to the results. The positive likelihood ratio was 2.41 (95% CI 2.28 to 2.52) which suggests that FC testing only slightly increases the probability of IBD disease. The negative likelihood ratio was 0.12 (95% CI 0.07 to 0.19) which suggests strong evidence to rule out IBD.
Abstract: IBS should be explained for the first time.	Thank you, we have added the explanation at the first mention of IBS.

Introduction: Given the aim of the study, it is important to indicate the prevalence of inflammatory bowel disease (IBD) in UK and in the world.	We have added the following paragraph on IBD prevalence in the UK and worldwide to the introduction. Prevalence of IBD is rising worldwide with highest reported prevalence values in North America (568 per 100,000 persons) and Europe (827 per 100,000 persons). ¹ UK prevalence estimates were 970/100,000 in 2017.²
---	---

VERSION 2 – REVIEW

REVIEWER	Joaquín Cubiella Hospital Universitario de Ourense, Ourense, Spain.
REVIEW RETURNED	14-Nov-2020
GENERAL COMMENTS	None, all my previous comments have been fully answered. Thank you.